# Position- and Hippo signaling-dependent plasticity during lineage segregation in the early mouse embryo

Eszter Posfai[1], Sophie Petropoulos[2,3,4], Flavia Regina Oliveira de Barros[1], John Paul Schell[2,4], Igor Jurisica[5,6,7], Rickard Sandberg[3,8], Fredrik Lanner[2,4], Janet Rossant[1,9]*

[1]Program in Developmental and Stem Cell Biology, Hospital for Sick Children, Toronto, Canada; [2]Department of Clinical Science, Intervention and Technology, Karolinska Institutet, Stockholm, Sweden; [3]Ludwig Institute for Cancer Research, Karolinska Institutet, Stockholm, Sweden; [4]Division of Obstetrics and Gynecology, Karolinska Universitetssjukhuset, Stockholm, Sweden; [5]Princess Margaret Cancer Centre, University Health Network, University of Toronto, Toronto, Canada; [6]Departments of Medical Biophysics and Computer Science, University of Toronto, Toronto, Canada; [7]Institute of Neuroimmunology, Slovak Academy of Sciences, Bratislava, Slovakia; [8]Department of Cell and Molecular Biology, Karolinska Institutet, Stockholm, Sweden; [9]Department of Molecular Genetics, University of Toronto, Toronto, Canada

*For correspondence: janet. rossant@sickkids.ca

Competing interests: The authors declare that no competing interests exist.

**Abstract** The segregation of the trophectoderm (TE) from the inner cell mass (ICM) in the mouse blastocyst is determined by position-dependent Hippo signaling. However, the window of responsiveness to Hippo signaling, the exact timing of lineage commitment and the overall relationship between cell commitment and global gene expression changes are still unclear. Single-cell RNA sequencing during lineage segregation revealed that the TE transcriptional profile stabilizes earlier than the ICM and prior to blastocyst formation. Using quantitative Cdx2-eGFP expression as a readout of Hippo signaling activity, we assessed the experimental potential of individual blastomeres based on their level of Cdx2-eGFP expression and correlated potential with gene expression dynamics. We find that TE specification and commitment coincide and occur at the time of transcriptional stabilization, whereas ICM cells still retain the ability to regenerate TE up to the early blastocyst stage. Plasticity of both lineages is coincident with their window of sensitivity to Hippo signaling.

## Introduction

The first two lineages to segregate during mammalian development are the inner cell mass (ICM) and the trophectoderm (TE). The TE is an extraembryonic tissue, giving rise to the trophoblast lineages of the placenta. The ICM will form two additional lineages before implantation - the pluripotent epiblast (EPI), giving rise to all germ layers of the embryo, and the primitive endoderm (PE), largely forming the endoderm layers of the yolk sacs (*Cockburn and Rossant, 2010*). At the blastocyst stage, the TE forms a monolayer tight junction-coupled epithelium enclosing the blastocoelic cavity, at one end of which lies the ICM. Inside and outside cell populations first form following the 8 to 16 cell divisions (*Anani et al., 2014*; *Dietrich and Hiiragi, 2007*; *Watanabe et al., 2014*). During the 16 cell stage and during the 16 to 32 cell divisions, division-independent and dependent cell

**eLife digest** In female mammals, conception is a complex process that involves several stages. First, an egg is released from the ovary and travels along a tube called the oviduct, where sperm from a male may fertilize it. If the egg is fertilized, the newly formed embryo moves into the womb, where it will then implant into the walls. In mice, it takes around four days for the embryo to implant and during this time, the cells in the embryo divide several times and start to specialize to form distinct cell types called lineages.

The first two lineages to form are known as the inner cell mass and the trophectoderm. The inner cell mass forms a ball of cells within the embryo and contains the precursors of all cells that build the animal's body. The trophectoderm forms a layer that surrounds the inner cell mass and will become part of the placenta (the organ that supplies the embryo with nutrients while it is in the womb). The embryo can organize these lineages without any instructions from the mother. However, it is still not clear when the cells start to differ from each other, and when they 'commit' to stay in these lineages.

Cells in the inner cell mass and trophectoderm have different gene expression profiles, meaning that many genes display different levels of activity in these two lineages. Posfai et al. use a technique called single-cell RNA sequencing to analyse gene activity as the inner cell mass and trophectoderm form in mouse embryos. By measuring changes in gene activity, it is possible to track their development and show which genes change expression levels when each lineage specifies and commits.

The experiments reveal that the inner cell mass and trophectoderm lineages develop at different times. As the inner cell mass forms, cells adopt the inner cell mass 'identity' before they commit to remaining in this lineage, revealing a window of time where different signals could still change the fate of the cells. However, when the early trophectoderm cells show the first signs of specialization, they also commit to their new identity at the same time.

These findings suggest that the different timings at which these cell lineages form might provide embryos with the means to organize their own cells. An important future challenge is to understand exactly how the cells commit to their fate.

internalization leads to dynamic morphological rearrangements (*Anani et al., 2014*; *Maître et al., 2016*; *Morris et al., 2010*; *Samarage et al., 2015*; *Watanabe et al., 2014*; *Yamanaka et al., 2010*). From the 32 cell stage onwards, apart from the relatively rare event of division-independent cell internalization, inside and outside positioning is generally a good indicator of ICM and TE lineage fates, respectively (*McDole et al., 2011*; *McDole and Zheng, 2012*; *Pedersen et al., 1986*; *Watanabe et al., 2014*; *Toyooka et al., 2016*).

During the 8 to 16 cell divisions, cells inherit varying amounts of apically localized proteins from the apical domain – the polarized outside surface forming at the 8 cell stage (*Anani et al., 2014*; *Johnson and Ziomek, 1981*; *Korotkevich et al., 2017*; *Watanabe et al., 2014*). Inside cells are apolar, while outside cells can either be apolar or polar. Interestingly, outside apolar cells were identified as the cells that internalize during the 16 cell stage in a division-independent manner (*Anani et al., 2014*; *Maître et al., 2016*). Although there is some evidence that individual blastomeres show variation in gene expression and epigenetic marks prior to the 8 cell stage, and that these differences may bias their future fate (*Biase et al., 2014*; *Burton et al., 2013*; *Goolam et al., 2016*; *Plachta et al., 2011*; *Torres-Padilla et al., 2007*; *White et al., 2016*), it is also clear that polarity differences are key to final assignment of cell fate. Polarity differences result in differential activation of the Hippo signaling pathway: active Hippo signaling in apolar cells sequesters the transcriptional co-activator Yap into the cytoplasm, while inactive Hippo in polar cells allows nuclear accumulation of Yap, and its interaction with the transcription factor Tead4 (*Hirate et al., 2013*; *Nishioka et al., 2009*). Nuclear Yap/Tead4 complexes are required for TE formation and are upstream regulators of Cdx2, a key TE-specific transcription factor (*Kaneko and DePamphilis, 2013*; *Nishioka et al., 2009*, *2008*; *Rayon et al., 2014*; *Yagi et al., 2007*). Activation of Cdx2 expression in TE progenitors leads to downregulation of the pluripotent factors, Oct4 and Nanog (*Chen et al., 2009*; *Niwa et al., 2005*; *Strumpf et al., 2005*), while in ICM progenitors Hippo signaling leads to upregulation of

Sox2, a co-factor with Oct4 in pluripotency (*Wicklow et al., 2014*). Thus differential Hippo activity is a key driver of ICM-TE lineage segregation. However, the exact time when differential Hippo signaling is instructive to establish cell fate is not known.

Moreover, the timing of cell fate commitment to ICM or TE is also not fully understood. A number of early studies attempted to define this timing by isolating inside and outside cells at different stages of development and then assessing cell potential in chimeras or re-aggregated embryos. However, these experiments yielded conflicting results. For example, inner cells were reported to have lost TE-forming potential by the 32 cell stage (*Tarkowski et al., 2010*), the early-mid blastocyst (~64 cell) stage (*Handyside, 1978*; *Rossant and Lis, 1979*; *Stephenson et al., 2010*; *Suwińska et al., 2008*) or the late blastocyst stage (*Grabarek et al., 2012*; *Hogan and Tilly, 1978*; *Spindle, 1978*). Outside cells were found to retain some plasticity up to the 32 cell stage (*Rossant and Vijh, 1980*; *Tarkowski et al., 2010*) but lost ICM potential when cavitation occurred during 32 cell stage (*Suwińska et al., 2008*). Using inside/outside position during cleavage as a marker for ICM and TE progenitors cannot be entirely accurate, given the dynamic rearrangements that take place, and so caution is needed in interpreting the conclusions of most of these experiments. This has led to considerable discrepancy between studies regarding the timing of restriction of developmental potential of the ICM.

In this study, we use single-cell RNA sequencing to reveal the temporal dynamics of gene expression during lineage segregation and identify known, as well as novel, markers of the process. We show that quantitative Cdx2-eGFP protein expression is an accurate readout of Hippo signaling activity and thus of the process of ICM-TE specification. We could then experimentally assess the potential of individual blastomeres scored for their level of Cdx2-eGFP expression and correlate with single-cell transcriptional profiles. We were able to resolve discrepancies in the literature and provide novel insights into the dynamics of lineage segregation.

## Results

### Cdx2-eGFP is an early readout of the Hippo signaling environment and the developing TE lineage

Using a Cdx2-eGFP fusion knock-in mouse line (*McDole and Zheng, 2012*) we confirmed that eGFP expression faithfully mimics endogenous Cdx2 expression dynamics (*Figure 1A*). Expression was first clearly detectable in early 16 cell embryos and gradually became restricted to the outer layer of the TE. We found that eGFP and endogenous Cdx2 fluorescent signals in *Cdx2-eGFP* heterozygous embryos showed a significant correlation from the 16 cell stage onwards.

To relate Cdx2-eGFP levels to cell position - used in previous studies to sort ICM and TE progenitors - we quantified eGFP in inside and outside cells (*Figure 1B*) in carefully staged embryos at different developmental times (*Figure 1—figure supplement 1*). We found that even from the early 16 cell stage onward, inside cells expressed on average significantly lower eGFP levels than outside cells. However, initially some inside and outside cells had overlapping eGFP levels, which gradually segregated by the late 32 cell stage.

Cdx2 expression is initiated in a heterogeneous, Tead4-independent manner at the morula stage, whilst later expression requires nuclear Yap/Tead4 activity in TE progenitor cells (*Nishioka et al., 2009*, *2008*). Yap localization is in turn regulated by Hippo signaling in the preimplantation embryo (*Cockburn et al., 2013*; *Nishioka et al., 2009*). To visualize how Hippo signaling differences take control of Cdx2 expression, we correlated nuclear/cytoplasmic (n/c) Yap ratios and Cdx2-eGFP levels in embryos at different stages (*Figure 1C*). We found that as soon as shuttling of nuclear Yap to the cytoplasm was initiated at the early 16 cell stage, Cdx2-eGFP levels started to show a positive correlation with n/c Yap ratios. This positive correlation progressively increased up to the early 32 cell stage. Thus, emerging Hippo signaling differences, starting at the early 16 cell stage, rapidly seize control of Cdx2 expression and up-regulation in TE progenitors is layered over initial heterogeneous, Tead4-independent signals.

At the 16 cell stage differential Hippo signaling has been shown to be dictated by differences in cell polarity, rather then position per se (*Anani et al., 2014*; *Maître et al., 2016*). While most outside cells are polarized and have nuclear Yap/Tead (inactive Hippo signaling), a population of apolar outside cells with cytoplasmic Yap (active Hippo signaling) has been reported. Moreover apolar outside

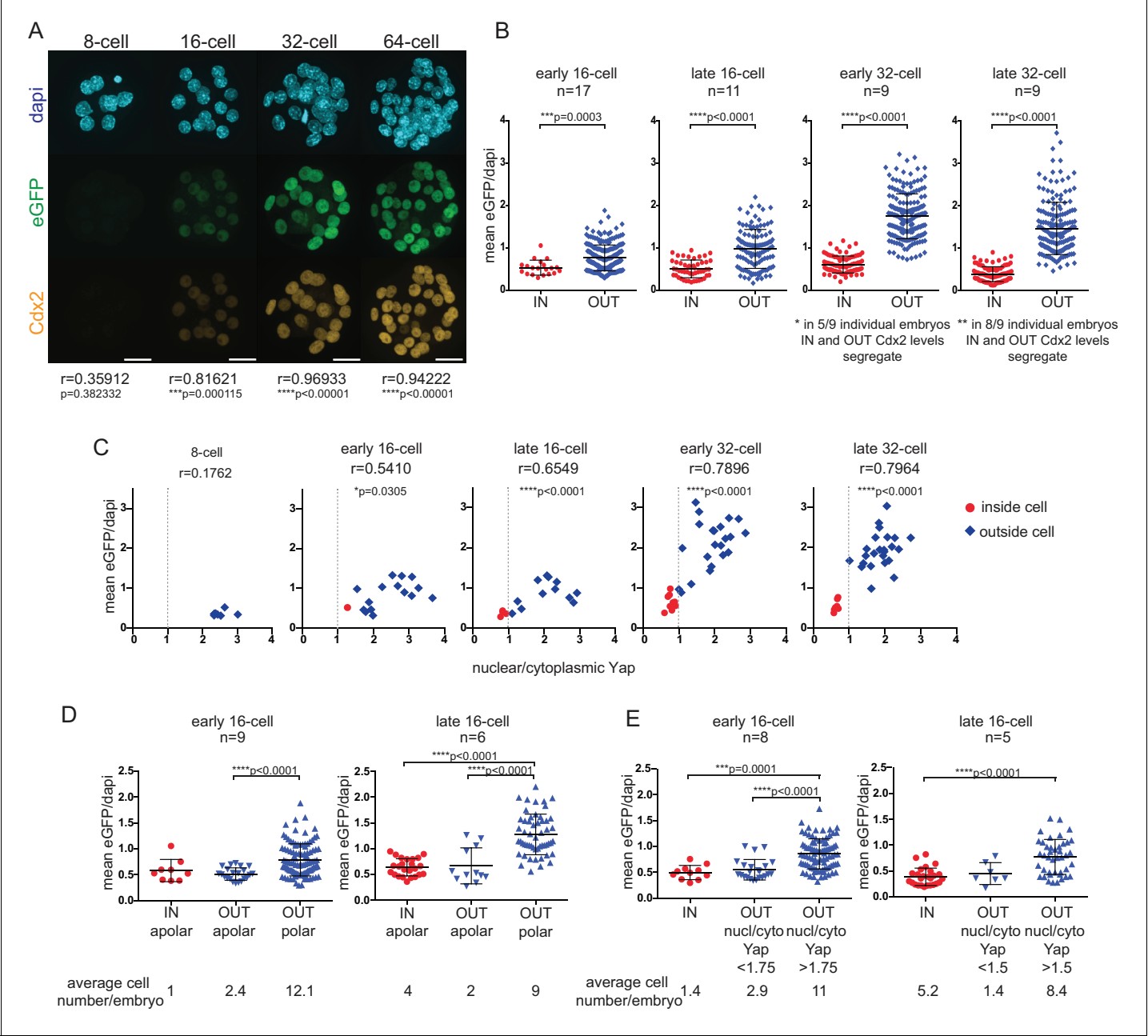

**Figure 1.** Cdx2-eGFP is an early marker of the developing TE lineage, governed by Hippo signaling differences from the early 16 cell stage. (A) Immunofluorescence staining against Cdx2 and eGFP in *Cdx2-eGFP* heterozygous embryos at different stages. Representative images of 10 8 cell, 39 16 cell, 35 32 cell and 11 64 cell embryos stained and imaged in two independent experiments. Scale bar: 25 μm. Correlation between eGFP and endogenous Cdx2 signals was calculated by measuring fluorescence intensities in individual cell nuclei and performing Pearson's correlation (r indicates coefficient). *p*-values are also given for each embryonic stage. (B) Mean fluorescence intensity of eGFP (Y-axis) in individual inside and outside cell nuclei of different stage *Cdx2-eGFP* embryos. Position was determined by co-staining embryos with phalloidin (F-actin) and cells with any surface membrane exposure were classified as outside. n indicates number of embryos. * and ** note how eGFP/Dapi measurements segregate in individual embryos. Statistical significance was calculated by Mann-Whitney test and significant *p*-values are indicated. Error bars: s.d. of mean. (C) Mean eGFP intensity relative to nuclear/cytoplasmic Yap ratio in individual inside (red) and outside (blue) cells in *Cdx2-eGFP* embryos at different stages. Representative measurements from 5 8 cell, 8 early 16 cell, 5 late 16 cell, 5 early 32 cell and 4 late 32 cell embryos are shown. All embryos were stained and imaged in one experiment. Correlation was calculated using Pearson's correlation (r indicates correlation coefficient) and *p*-value is given. (D–E) Mean fluorescenceintensity of eGFP (Y-axis) in single cells in different cell populations, in early and late 16 cell stage *Cdx2-eGFP* embryos. (D) Inside apolar, outside apolar and outside polar cell populations. (E) Inside cells, outside cells with low nuclear/cytoplasmic Yap ratio and outside cells with

*Figure 1 continued on next page*

*Figure 1 continued*

high nuclear/cytoplasmic Yap ratio. Polarity was determined by phospho-ezrin staining. n indicates number of embryos analyzed. Statistical significance was calculated by Kruskal-Wallis test and significant *p*-values are indicated. Error bars: s.d. of mean. Cells in M-phase are not included.

The following figure supplement is available for figure 1:

**Figure supplement 1.** Developmental staging of *Cdx2-eGFP* embryos.

cells have been shown to be ICM progenitors, which will eventually contribute to the inside compartment (*Anani et al., 2014*; *Maître et al., 2016*). To examine Cdx2-eGFP levels in different outside cell populations at the 16 cell stage we co-stained embryos with a polarity marker (*Figure 1D*) and found that apolar outside cells express low eGFP, similar to levels measured in inside cells. Similar results were obtained when n/c Yap ratios were used to distinguish between different outside cell populations (*Figure 1E*). These results indicate that upregulation of Cdx2-eGFP is downstream of polarity-induced Hippo inactivation (nuclear Yap localization), rather than position per se at the 16 cell stage.

Overall, this suggests that Cdx2-eGFP expression level, which is a downstream readout of nuclear Yap/Tead4, rather than position is the more appropriate way to sort putative ICM and TE progenitor cells at different stages of development.

## Single-cell RNA sequencing reveals gene expression dynamics in developing ICM and TE lineages

To explore the molecular dynamics underlying lineage segregation we performed single-cell RNA sequencing of individual cells isolated from *Cdx2-eGFP* embryos ranging from early 16 cell to 64 cell stages (*Figure 2A*). Cdx2-eGFP protein levels were measured in each cell prior to sequencing using quantitative fluorescence microscopy. After quality control, we retained 262 single-cell transcriptomes (70 early 16 cells, 43 late 16 cells, 49 early 32 cells, 39 late 32 cell and 61 64 cells) from 36 embryos, with an average of 7267 expressed genes per cell (RPKM > 1; Spearman pair-wise sample correlation $\geq$ 0.8). To examine how cells cluster with each other in an unbiased manner, we performed Principal Component Analysis (PCA) using the top 100 most variable genes across all cells (*Figure 2B–D and F*). The primary factor segregating cells was developmental time along PC1, where a clear progression towards two different cell populations was observed (*Figure 2B*). PC2 was strongly associated with known lineage markers – such as *Cdx2* mRNA (*Figure 2C*). We found that diversity among cells increased drastically between the late 16- and early 32 cell stages, suggesting that emergence of ICM and TE lineages at these stages.

We found a significant overall correlation (Pearson correlation coefficient r = 0.6605, p<0.0001) between *Cdx2* mRNA levels (*Figure 2C*) and Cdx2-eGFP protein levels (*Figure 2D*) in the same cell. Moreover, both align with the major transcriptional differences separating ICM and TE progenitors, further verifying Cdx2 expression levels as a means to read out the progression of lineage segregation.

To obtain ICM and TE progenitor-characteristic gene expression profiles we performed single-cell differential expression (SCDE) analysis (*Kharchenko et al., 2014*) between the two clearly distinct cell populations at the early 32 cell stage and found 135 ICM-specific and 207 TE-specific differentially expressed genes (DEGs) (*Figure 2—source data 1*). In order to assign lineage identity to all cells, we applied Spearman's rank correlation clustering using the top 50 genes for each lineage from the early 32 cell DEGs as input and visualized groupings (*Figure 2E*). We observed clear ICM and TE populations, as expected, at early 32-, late 32- and 64 cell stages. Interestingly, we found a subset of the 16 cell stage cells clustered with the ICM (33 out of 113 cells) and TE (24 out of 113 cells) groupings, whereas the remaining (56 out of 113 cells) where positioned in a third cluster categorized as 'co-expressing' (CO) (*Figure 2E*). This indicates that some cells at the 16 cell stage already initiated transcriptional linage divergence. Both Spearman's rank correlation clustering (*Figure 2E*) and PCA clustering (*Figure 2F*) revealed that the ICM and CO populations clustered closer together and further from the TE group indicating a closer relationship between ICM and CO cells.

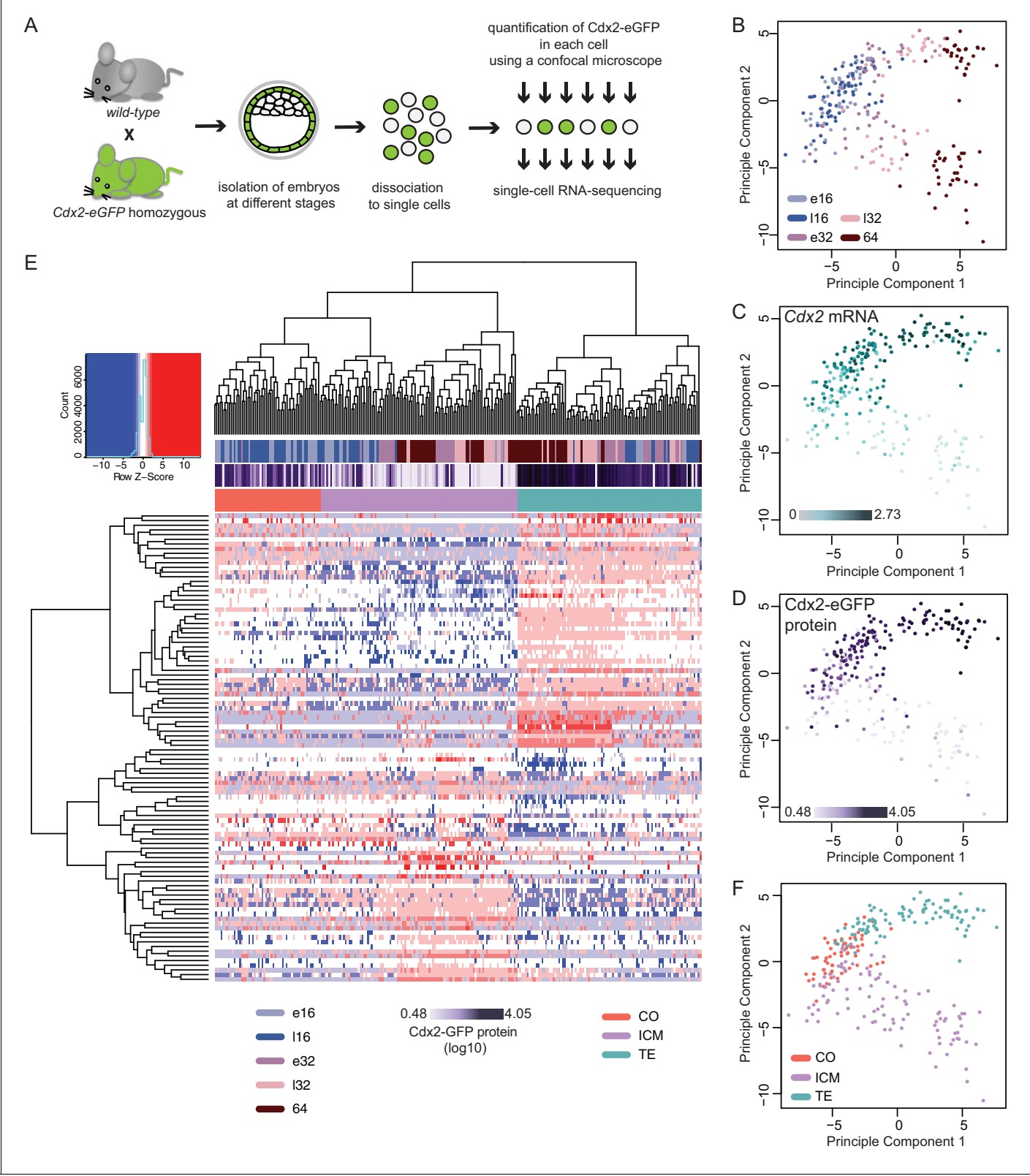

**Figure 2.** Single–cell RNA sequencing reveals gradual emergence of ICM and TE lineages. (**A**) Experimental outline for harvesting single cells for RNA sequencing. (**B–D**) Principal component analysis using top 100 variable genes across all cells, where each cell is annotated for (**B**) developmental time (**C**) corresponding expression level (log10 RPKM) of *Cdx2* mRNA (**D**) corresponding Cdx2-eGFP protein (measured prior to RNA sequencing). (**E**) Heatmap showing log10 RPKM expression level of early 32 cell lineage signature genes in all 262 cells. Cells were annotated for developmental time, *Figure 2 continued on next page*

*Figure 2 continued*

corresponding Cdx2-eGFP values and lineage identity, assigned based on Spearman's rank correlation clustering. (**F**) Principal component analysis using top 100 variable genes across all cells, showing TE, ICM and co-expressing (CO) lineage assignment of cells.

The following source data is available for figure 2:

**Source data 1.** Excel file of differentially expressed genes from SCDE analysis between 'Cdx2-low' and 'Cdx2-high' cell populations (based on PCA groupings) at the early 32 cell stage.

To assess transcriptional differences underlying lineage segregation we performed SCDE analysis between ICM, TE and CO groups at each developmental stage (*Figure 3A* and *Figure 3—source data 1*) and found an increasing difference between ICM and TE lineages with developmental time. At the 16 cell stage we found 55 DEGs between ICM and TE profiles, with a more extensive gene network (42 genes) expressed in TE cells. These included known TE markers *Cdx2* (*Figure 2B*) *Id2*, *Dppa1*, *Ptges*, *Krt8* and *Krt18* (*Figure 3C*) as well as genes previously not or less-associated with TE development, such as *Lrp2*, *Dab2* (*Figure 3C*), *Bmyc* and *Dusp4*. Thus the first transcriptional differences arising among lineage progenitors are the activation of TE-specific genes.

To assess whether TE-specific genes identified by our single-cell RNA sequencing are candidates for being direct targets of Yap/Tead4 activity, we compared previously published Tead4 ChIP sequencing data in trophoblast stem cells (*Home et al., 2012*) - *in vitro* derivatives of the TE lineage - with our TE-specific genes (*Figure 3D*). We found that 162 out of 404 TE-specific genes were associated with at least one Tead4 binding site (*p*-value<0.038; hypergeometric test). Additionally, we were able to detect protein expression of 4 early TE-specific genes (*Krt8*, *Krt18*, *Lrp2* and *Dab2*) in wild-type embryos, but found no or very little protein expression in *Tead4* maternal/zygotic mutant embryos (*Figure 3E*). All four genes were expressed in the TE as early as the 16 cell stage and were associated with Tead4 peaks in trophoblast stem cells. These findings identify *Krt8*, *Krt18*, *Lrp2* and *Dab2* as likely direct target genes of Yap/Tead4 activity in the early embryo.

To assess temporal gene expression dynamics within each lineage, we conducted SCDE analysis between different time points within ICM and TE lineages and between CO/ICM and CO/TE lineages (*Figure 3B* and *Figure 3—source data 2*). We found that the largest change in gene expression in both lineages occurred at the 16 to early 32 cell transition, when most CO cells specified either into ICM or TE profiles. It is here that we saw upregulation of known early markers in the ICM, such as *Sox2* and *Nanog* (*Figure 3—figure supplement 1*).

Within the ICM lineage we found 177 DEGs between the 16 cell/early 32 cell stages, 112 DEGs between early 32 cell/late 32 cell stages and only 22 DEGs between the late 32 cell/64 cell stages. In contrast, within the TE lineage a relatively mild maturation of the transcriptional profile was seen with time, suggesting a sharp specification event when CO profiles resolve into TE profiles at the 16 and early 32 cell stages. Only 21, 13, and 43 DEGs were found between 16 cell/early 32 cell, early 32 cell/late 32 cell and late 32 cell/64 cell TE, respectively. Examples of genes showing lineage and stage specific expression patterns are found in *Figure 3—figure supplement 1*.

## Developmental potential of single embryonic cells assayed by aggregation to host morula embryos

To correlate developmental potential of single cells with their transcriptional profiles, we tested the lineage contribution of cells with varying Cdx2-eGFP levels from different developmental stages in a morula aggregation assay. As outlined (*Figure 4A*), we harvested embryos at different stages from Cdx2-eGFP x CAG-DsRed (*Vintersten et al., 2004*) crosses, dissociated them to single cells and measured Cdx2-eGFP in individual cells. These individual donor cells were then aggregated to a wild-type host morula and resulting chimeras were cultured to embryonic day 4.5 (E4.5). Chimeras were immunostained for DsRed to visualise the progeny of the aggregated single donor cell and for ICM (Klf4) and TE (Cdx2) lineage markers (*Figure 4B*). Donor cells isolated from 8 cell embryos did not express Cdx2-eGFP and contributed to ICM, TE or both lineages in chimeras (*Figure 4C*). Some donor cells from early 16 cell embryos started to express Cdx2-eGFP, and a small, yet significant bias was detected of Cdx2-eGFP high cells contributing to the TE and Cdx2-eGFP low cells contributing to ICM. This bias progressively increased with the developmental stage of the donor cell. We

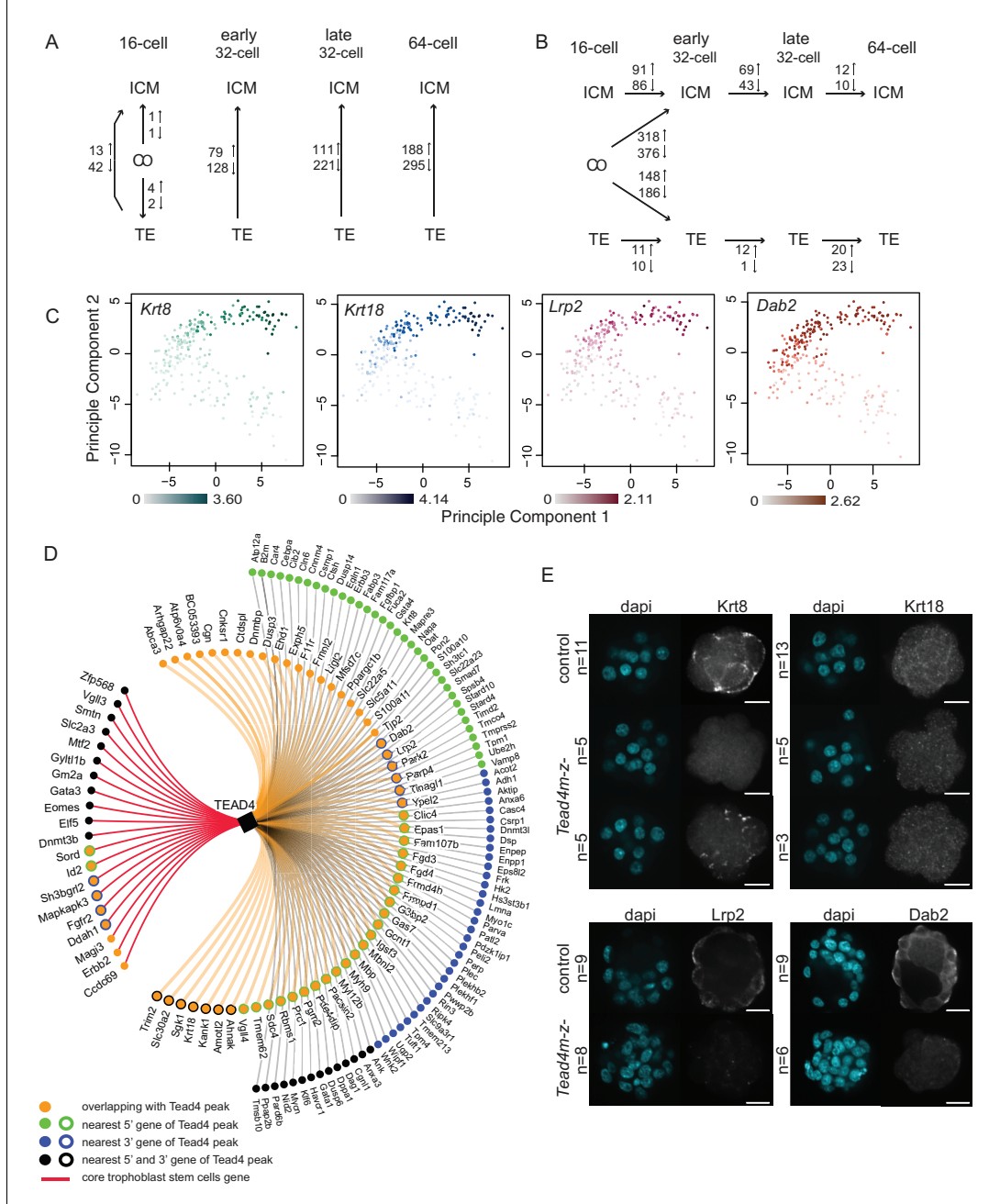

**Figure 3.** Different gene expression dynamics during development of ICM and TE lineages. (A–B) Summary of the number of genes differentially expressed from SCDE analysis (A) between lineages within each developmental time point and (B) between developmental time points within each lineage. Due to the low number of ICM and TE cells at the early and late 16 cell stages, these time points were pooled. (C) Principal component analysis using top 100 variable genes across all cells, annotated for the expression level (log10 RPKM) of early TE-associated genes *Krt8*, *Krt18*, *Lrp2* and *Dab2*. (D) TE-specific genes identified by single-cell RNA sequencing associated with at least one Tead4 binding site in trophoblast stem cells (***Home et al., 2012***). Gene association with Tead4 binding sites was defined as in Home *et al.* – genes overlapping with Tead4 peaks and genes nearest to Tead4 peaks in 5' and 3' directions are considered. Core trophoblast genes (***Ralston et al., 2010***) are also shown (red line). (E) Representative immunofluorescence stainings of control (*Tead4m-z+*) and *Tead4* maternal/zygotic mutant (*Tead4m-z-*) embryos for Krt8 and Krt18 (16 cell stage embryos) and Lrp2 and Dab2 (32 to 64 cell stage embryos; Lrp2 and Dab2 were not detected in earlier stage embryos). n indicates total number of embryos analyzed. Scale bar: 25 μm.

The following source data and figure supplements are available for figure 3:

**Source data 1.** Excel file of differentially expressed genes from SCDE analysis between lineages within each developmental time point.

*Figure 3 continued on next page*

*Figure 3 continued*
**Source data 2.** Excel file of differentially expressed genes from SCDE analysis between developmental time points within each lineage.
**Figure supplement 1.** Examples of lineage and stage specific gene expression patterns identified by single-cell RNA sequencing.
**Figure supplement 2.** ICM commitment coincides with initiation of epiblast and primitive endoderm segregation.

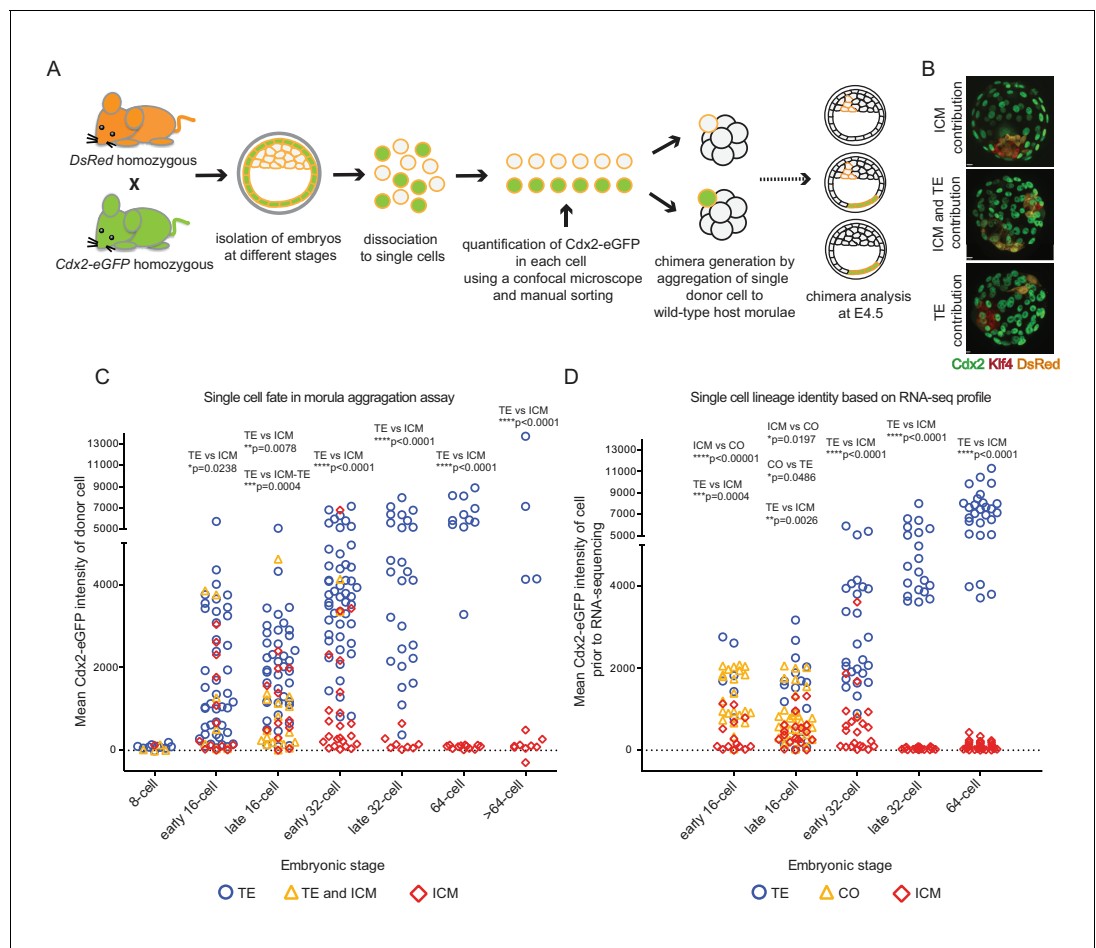

**Figure 4.** Individual cells with different Cdx2-eGFP levels aggregated to host morulae show gradual and differential loss of developmental potential over time. (A) Experimental outline of morula aggregation assay. (B) Examples of chimeras with TE, ICM and TE and ICM contributions, analyzed at E4.5 by immunofluorescence staining. (C) Plot showing all aggregation chimera results. Each data point represents a single donor cell isolated from different stage embryos (X axis) with the level of mean Cdx2-eGFP measured in each cell before aggregation (Y axis). Donor cells are color coded for the lineage their progeny contributed to in the chimera. Statistically significant differences in Cdx2-eGFP intensities between different contributions (ICM vs TE and TE vs ICM-TE) were calculated using Mann-Whitney test and significant *p*-values are indicated. (D) Plot showing lineage identities assigned to single cells based on RNA sequencing profiles. Each data point represents a single cell isolated from different stage embryos (X axis) with the level of mean Cdx2-eGFP measured in each cell before sequencing (Y axis). Cells are color coded according to their lineage profiles. Statistically significant differences in Cdx2-eGFP intensities between different lineage groups (ICM vs TE, TE vs CO and ICM vs CO) were calculated using Mann-Whitney test and significant *p*-values are indicated.
The following figure supplement is available for figure 4:

**Figure supplement 1.** Live imaging chimera formation following single early 32 cell stage donor cell aggregation to host morula.

also noted that the ability of a single cell to give rise to both lineages sharply decreased at the 16 to 32 cell transition. Interestingly, Cdx2-eGFP low cells from the early 32 cell stage exclusively contributed to the ICM, while it took until the late-32 cell stage for Cdx2-eGFP high cells give rise to solely TE, thus revealing a different time line of fate restriction of Cdx2-eGFP low and high cells.

Remarkably, when ICM, TE or CO cells, annotated based on RNA sequencing profiles, were plotted in a similar manner to lineage contribution results from the morula aggregation experiments, the plots were strikingly similar (*Figure 4C and D*). This may suggest that cell behavior in the morula aggregation assay reads out the gradual process of ICM and TE specification as judged by dynamic transcriptional profiles.

### Live imaging of chimera formation reveals dynamic behavior of cells during aggregation and lineage development

We observed the same fate outcomes of donor cells isolated from early 32 cell stage embryos in live imaging aggregation experiments as in our end-point analysis. All Cdx2-eGFP low donor cells imaged (n = 5) (*Video 1*; *Figure 4—figure supplement 1*) fully internalized in chimeras between 8 to 10 hr after aggregation, with Cdx2-eGFP levels remaining low throughout and donor cells dividing only after taking up an ICM position. In majority of chimeras with Cdx2-eGFP high donor cells (n = 4/6), cells remained on the outside, maintained high Cdx2-eGFP expression and underwent division with all progeny also remaining in TE position (*Video 2*; *Figure 4—figure supplement 1*). In remaining chimeras with Cdx2-eGFP high donor cells (n = 2/6), cells were internalized while in the process of downregulating Cdx2 and finally divided in an ICM position (*Video 3*; *Figure 4—figure supplement 1*).

We thus found that division was not a driving force of cell internalization and that complete downregulation of Cdx2 is not a prerequisite for internalization. This supports previous observations that some Cdx2 positive cells can internalize even in intact embryos at the 32 cell stage, downregulate Cdx2 and integrate into the ICM (*McDole and Zheng, 2012*; *Toyooka et al., 2016*). It further suggested that the morula aggregation assay read out the state of specification of the isolated blastomeres and not necessarily their full developmental potential.

### Developmental potential assayed by reconstructing embryos from cells sorted by Cdx2-eGFP expression levels

Next we tested the developmental potential of cells using a different assay, in which we reconstructed entire embryos from cells sorted by levels of Cdx2-eGFP expression (*Figure 5A*), thus removing the influence of the host embryo environment. At each developmental stage examined, the appropriate number of Cdx2-eGFP low or Cdx2-eGFP high cells, or a random mixture of cells was re-aggregated in groups. Such re-constructed embryos were cultured to E4.5 and analyzed for ICM (Sox2 and Gata4) and TE (Cdx2) lineage markers. We found that embryos reconstructed solely from cells with low or high Cdx2-eGFP levels isolated from late 16 cell embryos readily formed both ICM and TE lineages by E4.5 (*Figure 5B*; *Figure 5—figure supplement 1*). Similarly all embryos made from Cdx2-eGFP low and most embryos from Cdx2-eGFP high early 32 cell stages still recapitulated both ICM and TE lineages (*Figure 5C*; *Figure 5—figure supplement 1*), although we observed a non-significant decrease in the number of ICM cells formed in embryos from Cdx2-eGFP high cells compared to embryos made from Cdx2-eGFP low or random cells (Mann-Whitney test, *p*-value=ns). Embryos made from late 32 Cdx2-eGFP low cells still mostly reconstituted both ICM and TE lineages, while Cdx2-eGFP high cells by this stage completely lost their ability to form ICM and only produced Cdx2 positive TE (*Figure 5D*; *Figure 5—figure supplement 1*). It was only by the 64 cell stage that the majority of Cdx2-eGFP low

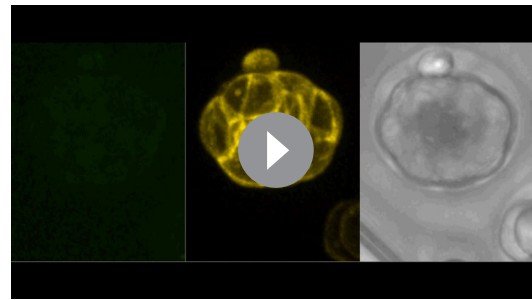

**Video 1.** Live imaging a single Cdx2-eGFP low donor cell from a 32 cell stage embryo aggregating with a host morula - donor cell moves in and contributes to the ICM.      Related to *Figure 4*.

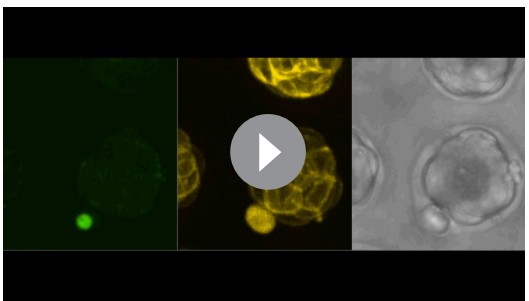

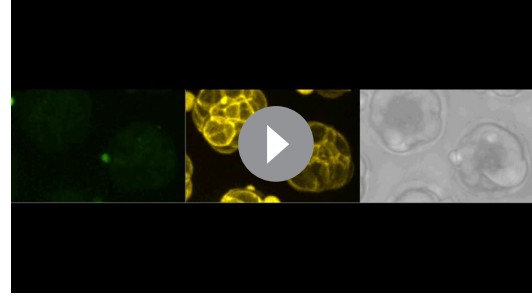

**Video 2.** Live imaging a single Cdx2-eGFP high donor cell from a 32 cell stage embryo aggregating with a host morula – donor cell stays on the surface and contributes to the TE.      Related to *Figure 4*.

**Video 3.** Live imaging a single Cdx2-eGFP high donor cell from a 32 cell stage embryo aggregating with a host morula - donor cell moves in and contributes to the ICM.      Related to *Figure 4*.

cells lost their ability to make TE (*Figure 5E*; *Figure 5—figure supplement 1*). Out of eight embryos made from Cdx2-eGFP low cells, only three showed re-expression of Cdx2 in a few cells (4, 6 and 3 cells per embryo), two without cavitating and one with a tiny cavity. A fourth embryo developed a small cavity but without any re-expression of Cdx2. All other embryos (n = 4/8) remained as tightly packed clusters without Cdx2 re-expression or cavitation.

Interestingly, this assay revealed a different timeline for fate restriction than the morula aggregation assay. Cdx2-eGFP high cells completely lost their ICM forming ability by the late 32 cell stage, while Cdx2-eGFP low cells retained TE potential at late 32 cell stage and did not lose TE potential until the 64 cell stage.

## Fate restriction as assessed by re-aggregation assay coincides with cells becoming refractory to Hippo signaling-induced changes

Differential Hippo signaling plays a key role in ICM-TE lineage specification (*Cockburn et al., 2013*; *Hirate et al., 2013*; *Lorthongpanich et al., 2013*; *Nishioka et al., 2009*). However, the exact period over which Hippo signaling can influence cell fate has not been addressed. We used an inhibitor of Rho-associated protein kinase (ROCKi) and inducible expression of dominant negative Lats2 to activate or block Hippo signaling, respectively at different times during development to address this issue.

Treatment of embryos from the 2 cell stage on with ROCKi was shown to enhance ICM and suppress TE characteristics through activation of Hippo signaling (*Kono et al., 2014*). We treated embryos with ROCKi for a 24 hr period, starting at different stages (*Figure 6A*), and analyzed Cdx2 expression following treatment as a measure of cell fate restriction. We found that ROCKi treatment beginning at the 16 cell stage significantly reduced the percent of Cdx2 positive cells compared to controls (35% and 64%, respectively) (*Figure 6B*). We confirmed that this coincided with ectopic Hippo activation as assessed by loss of nuclear YAP and an increase in phosphorylated cytoplasmic Yap (*Figure 6—figure supplement 1*). When embryos began treatment at stages later than the 16 cell stage, there was a gradual resistance to the effect of ROCKi, with no effect of activating Hippo signaling on Cdx2 expression by the late 32 cell stage.

For the reverse experiment, to determine when ICM progenitors cease responding to inactivation of Hippo signaling, we employed an inducible genetic approach. Transgenic embryos were generated with a doxycycline (Dox)-inducible dominant-negative Lats2 (DN Lats2) tagged with mCherry (*Figure 6C*) and induced at different stages for 24 hr. Following treatment, embryos expressing high levels of mCherry were analyzed for ICM (Sox2 and Sox17) and TE (Cdx2) lineage markers (*Figure 6D and E*). Our approach to generate transgenic embryos typically resulted in mosaic integration of inducible DN Lats2-mCherry, which allowed us to analyze both transgenic (mCherry positive) and non-transgenic (mCherry negative) ICM cells within the same embryo. Up to the early 32 cell stage, DN Lats2 expression in inner cells could induce fate change as indicated by gain of Cdx2 and loss of ICM markers in most mCherry positive inner cells. At the late 32 cell stage we observed some mCherry positive inner cells (14%) co-expressing Cdx2 and ICM markers, while majority (80%)

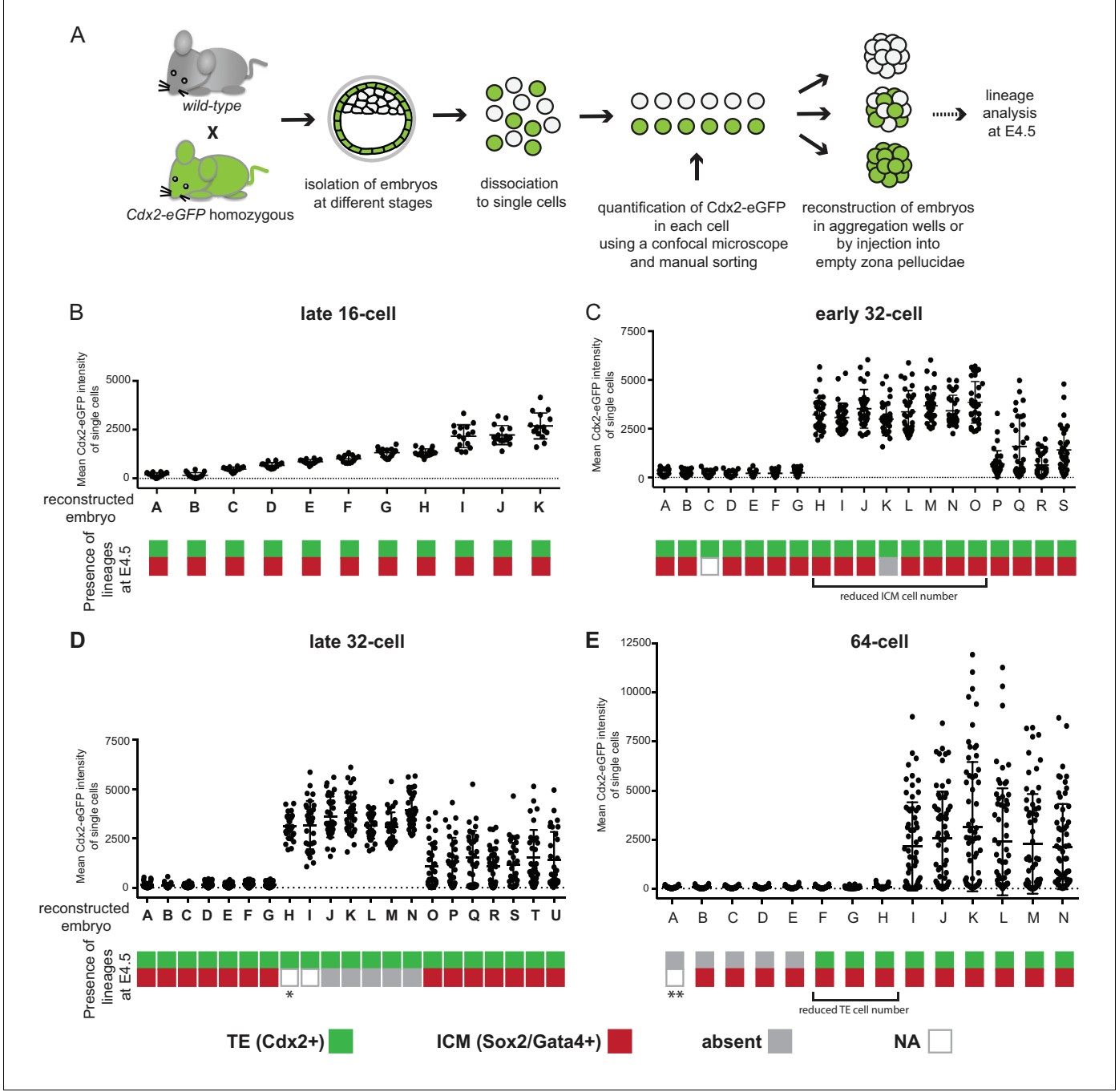

**Figure 5.** Embryos reconstructed entirely from Cdx2-eGFP low or high cells loose their potential to recapitulate ICM and TE lineages at different times. (A) Experimental outline to reconstruct embryos entirely of Cdx2-eGFP low, high or random cells. (B–E) Plots showing embryo reconstructions from single cells isolated from (B) late 16 cell, (C) early 32 cell, (D) late 32 cell and (E) 64 cell stages. Each embryo (X axis, labeled with letters) was reconstructed from Cdx2-eGFP-quantified (Y axis) single cells. Color-coding below indicates the presence of Cdx2 positive TE (green) and Sox2 or Gata4 positive ICM (red) cells in reconstructed embryos at E4.5. Grey indicates the absence of a lineage; white (N/A) indicates the embryo was lost during immunofluorescence staining, thus information is only available of the TE lineage from the live Cdx2-eGFP marker before fixation. * embryo visually only consisting of trophoblast vesicles. ** embryo morphology like B-E embryos, likely contains both Gata4 and Sox2 positive cells.

The following figure supplement is available for figure 5:

**Figure supplement 1.** Representative images of embryo reconstructions from single cells at different stages.

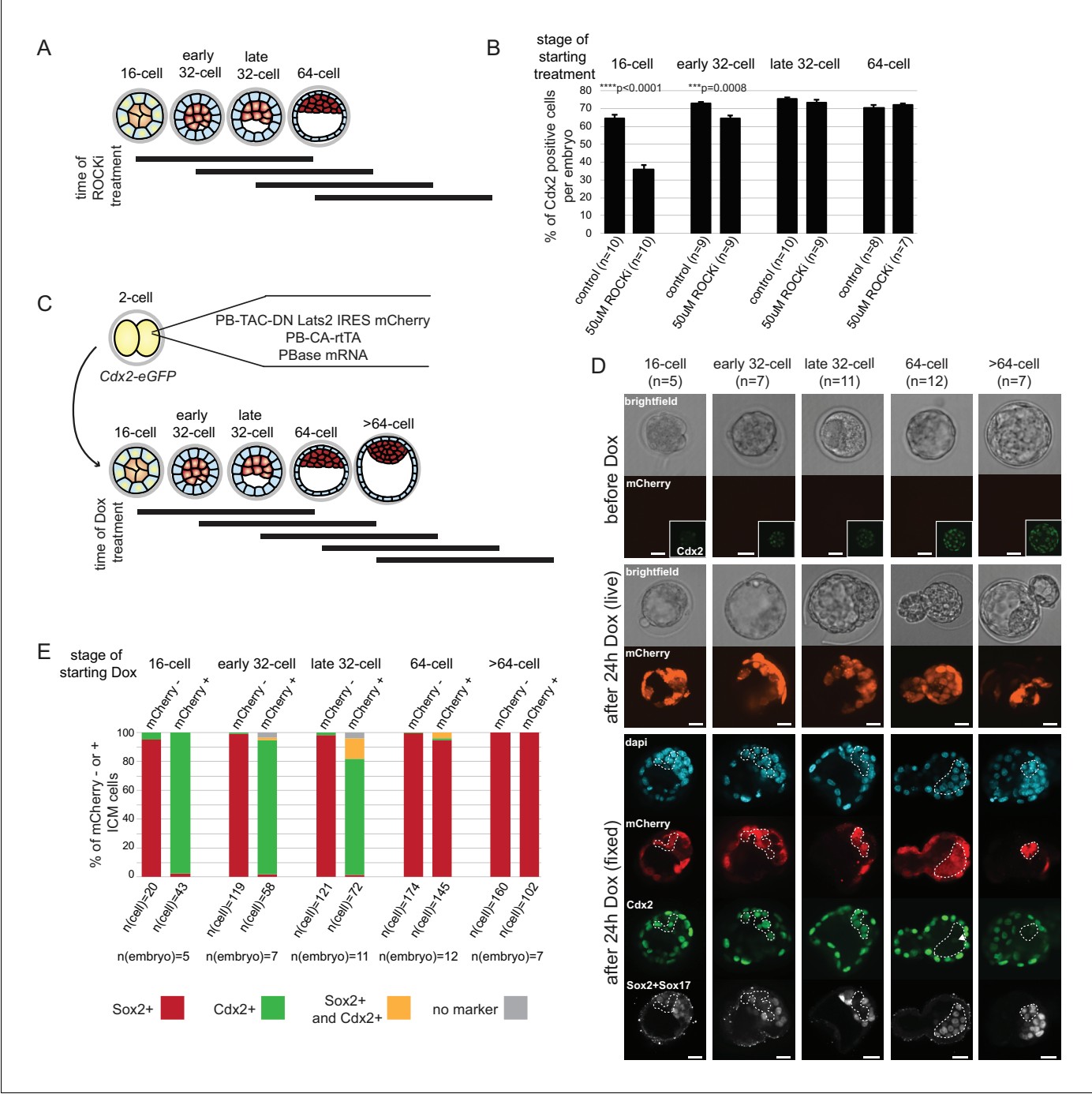

**Figure 6.** ICM and TE progenitors show loss of responsiveness to Hippo signaling manipulation at the same time as they loose responsiveness to positional changes. (A) Overview of Hippo signaling activation time course. Each bar represents 24 hr of 50 μM ROCKi treatment. (B) Percent of Cdx2 positive cells per embryo cultured for 24 hr in control or ROCKi conditions. Label on top indicates the stage embryos started treatment. n indicates number of embryos analyzed. Statistical significance was calculated using *t*-test and significant *p*-values are indicated. Error bars: s.d. of mean. (C) Strategy for inducible Hippo signaling inactivation. Mostly mosaic Dox-inducible DN Lats2-IRES-mCherry transgenic embryos were generated. Each bar represents 24 hr of Dox treatment. (D) Dox-inducible DN Lats2-IRES-mCherry transgenic embryos were imaged before Dox treatment (top panel) and the same embryo was imaged following 24 hr of Dox live (middle panel) and fixed/stained for lineage markers (bottom panel). A representative embryo is shown for each stage. Live mCherry is shown as an extended focus image, immunofluorescence stainings shown as single plane images. mCherry positive ICMs in mosaic transgenic embryos are circled with a dotted line. Arrow points to a rare ICM cell in a 64 cell stage-induced embryo with weak Cdx2 expression, which also co-expressed an ICM marker. Scale bar: 25 μm. n indicates number of transgenic embryos analyzed. (E) All mCherry negative (non-transgenic control) and mCherry positive (DN Lats2-mCherry transgenic) ICM cells were scored in mosaic embryos for presence or

*Figure 6 continued on next page*

*Figure 6 continued*

absence of lineage markers following 24 hr of Dox treatment by immunofluorescence staining. Cells with different lineage marker expression are shown as percent of all mCherry negative or mCherry positive ICM cells analyzed. n(cell) indicates number of cells analyzed at each stage and n(embryo) indicates number of embryos cells were pooled from. Chi-squared test was used to test whether cell fate was affected by DN Lats2-mCherry expression. 16 cell *p*-value=8.48491E-18; early 32 cell *p*-value=5.50841E-34; late 32 cell *p*-value=6.32116E-35; 64 cell *p*-value=0.004103716; >64 cell *p*-value=0.588416983.

The following figure supplements are available for figure 6:

**Figure supplement 1.** Effect of ROCKi treatment on cell number and Hippo signaling.

**Figure supplement 2.** Expression of mCherry only does not influence cell fate in the embryo.

of cells still fully converted to expressing only Cdx2. When DN Lats2 was induced at the 64 cell stage, however, most inner cells maintained ICM marker expression and did not re-express Cdx2. Only 1% of mCherry positive inner cells switched to expressing only Cdx2 and 4% expressed markers of both lineages. All inner cells in embryos induced at >64 cell stage showed commitment to ICM. Additionally, we noted that expression of only mCherry (without DN Lats2) did not influence cell fate at any stage (*Figure 6—figure supplement 2*).

These results are in good agreement with our findings from embryo reconstruction experiments, indicating that on a mechanistic level, loss of ICM potential of Cdx2-eGFP high cells and loss of TE potential of Cdx2-eGFP low cells corresponds to the time when cells become refractory to Hippo signaling activity or inactivity, respectively.

## Discussion

While existing expression profiling datasets provide only limited or partial coverage of ICM-TE lineage segregation (*Deng et al., 2014*; *Goolam et al., 2016*; *Graham et al., 2014*; *Guo et al., 2010*; *Ohnishi et al., 2014*; *Tang et al., 2011*), our study offers a large, comprehensive single-cell global transcriptional dataset spanning the entire window of lineage segregation with remarkable temporal resolution. We show a gradual separation of ICM and TE lineages starting at the 16 cell stage, where the first transcriptional changes to distinguish cell populations are the activation of TE-specific genes, including *Cdx2*. *Cdx2* is the only known downstream target of nuclear Yap/Tead4 in the embryo (*Rayon et al., 2014*). We now present evidence that a number of the additionally identified early TE genes, such as *Krt8*, *Krt18*, *Dab2* and *Lrp2* are likely targets of nuclear Yap/Tead4 as well. Whether these transcriptional differences at the 16 cell stage relate to earlier heterogeneities among blastomeres, as reported by other groups (*Biase et al., 2014*; *Burton et al., 2013*; *Goolam et al., 2016*; *Plachta et al., 2011*; *Torres-Padilla et al., 2007*; *White et al., 2016*), is not clear, although we did not observe correlated differential expression of genes previously suggested to be involved in generating such heterogeneities (e.g. *Carm1*, *Prdm14* or *Sox21*) (*Burton et al., 2013*; *Goolam et al., 2016*; *Torres-Padilla et al., 2007*). While we detected the initiation of lineage segregation in some cells at the 16 cell stage, others were still in a state of co-expression. The division to the early 32 cell stage marked a drastic transcriptional change, resolving the co-expressing state into ICM or TE profiles. Interestingly, we found that as the TE profile appeared, it also stabilized. On the other hand, the emerging ICM profile underwent considerable maturation until the 64 cell stage, at which point segregation into EPI and PE lineages is initiated (*Guo et al., 2010*; *Kurimoto et al., 2006*; *Ohnishi et al., 2014*); and in this study [*Figure 3—figure supplement 2*]).

Importantly, we link global gene expression patterns to functional measures of cell fate and potential, by charting different experimental readouts as a function of developmental time and level of Cdx2-eGFP expressed in individual cells. We experimentally tested the potential of single cells to contribute to developing lineages of the early embryo using a morula aggregation assay and found that cell behaviors observed in this test reflected the progress of lineage separation apparent from transcriptional profiling. We suggest that in this assay, a single cell finds itself in a competitive host environment and thus can act according to its intrinsic lineage identity: a specified TE progenitor will remain on the outside and contribute to the TE, while a specified ICM progenitor will move in and contribute to the ICM. Live imaging of chimera formation further supported such dynamic behaviors

of ICM and TE progenitor cells. Thus we propose that despite the experimental manipulations involved, the morula aggregation assay reports cell fate specification. In contrast, when an entire embryo is reconstructed from only ICM or only TE progenitor cells, competition with the host cells is removed and some cells are inevitably forced inside and others outside. As such, this assay reveals the full potential of cells to change fate when forced into different positions. In this assay, ICM cells showed an extended period of lineage plasticity when compared with the results of the morula aggregation assay. We further showed that loss of responsiveness towards Hippo signaling manipulation coincided with the timing of fate restriction in ICM and TE progenitors, as read out by the reconstruction assay. Thus in the intact embryo and even in the morula aggregation assay, transcriptional profiles predict TE and ICM fates, while in the reconstruction assay and the Hippo manipulation experiments, the full potential of cells to respond to positional changes and associated signaling environments is revealed.

Importantly, in previous studies only one type of assay was used - either morula aggregation or reconstruction (*Grabarek et al., 2012*; *Handyside, 1978*; *Hogan and Tilly, 1978*; *Rossant and Lis, 1979*; *Spindle, 1978*; *Stephenson et al., 2010*; *Suwińska et al., 2008*; *Tarkowski et al., 2010*). By performing these two assays side by side, we can observe that the morula aggregation assay largely tests cell fate, while the reconstruction assay reveals full developmental potential, thus reconciling differences reported among the previous studies.

When comparing the dynamics of fate restriction shown by cells in the different assays (summarized in *Figure 7*), we observed that TE progenitors exhibited similar timing of fate restriction in all assays employed. A majority of Cdx2-eGFP high cells at the early 32 cell stage produced only TE in all three assays, which also corresponded to the time by which most co-expression profiles resolved, as shown by RNA sequencing. On the other hand, ICM progenitors showed fate restriction earlier in the morula aggregation test (by the early 32 cell stage) than they did in the embryo reconstruction and Hippo-inactivation assays (by the 64 cell stage), revealing a time window between lineage specification and commitment. Correspondingly, gene expression dynamics in the ICM lineage also reflected this prolonged maturation.

Why should there be asynchronous lineage commitment in the developing embryo? During the 16 to 32 cell divisions a number of outside cells are internalized as a result of the cleavage plane orientation (*Morris et al., 2010*; *Yamanaka et al., 2010*). However, a recent study revealed that daughter cells pushed inwards during mitosis often sort back to the surface (*Watanabe et al., 2014*). We propose that the timely commitment of majority of TE cells that we observe between the late 16 and early 32 cell stages may be the driving force of this sorting process, allowing generation of a differentiated polarized epithelial layer that would stabilize the inside compartment. Cell

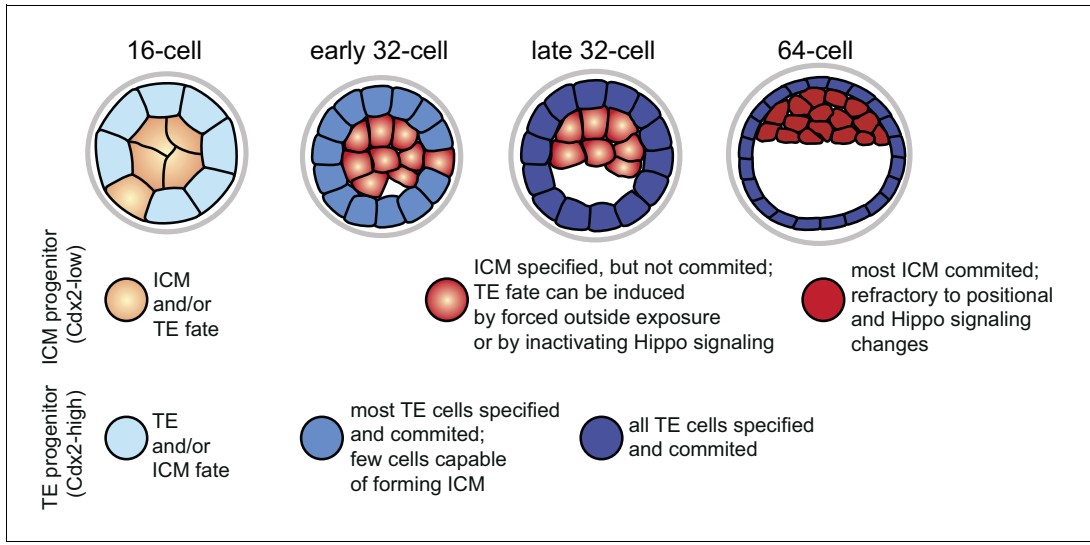

**Figure 7.** Graphical summary of specification and commitment of ICM and TE progenitors.

divisions do not cause spatial perturbation in the ICM, thus functional commitment to this lineage may not be needed at this stage. Instead, we find that the ICM loses the ability to become TE when the first heterogeneities in gene expression demarcating the start of the second lineage segregation arise. This is in line with an earlier study, which showed that outer cells of isolated ICMs of early blastocysts mostly formed trophoblast outgrowths, while isolated ICMs from later stage blastocysts mostly formed PE-like outgrowths (*Nichols and Gardner, 1984*). We suggest a possible functional relationship between loss of ICM plasticity and initiation of EPI and PE differentiation programs, which requires further investigation. In conclusion, we propose that asynchronous lineage commitment may be a mechanism contributing to the regulative nature of the preimplantation embryo, ensuring that the correct number of cells is allocated to inside and outside compartments.

## Materials and methods

### Mouse lines and embryos

ICR (Crl:CD1(ICR)) breeding stock from Charles River, Montreal, Canada), *Cdx2-eGFP* (knock-in fusion to endogenous locus) (RRID:IMSR_KOMP:VG12984) (*McDole and Zheng, 2012*), *DsRed* (RRID:IMSR_JAX:005441) (*Vintersten et al., 2004*), *Tead4 fl/fl* (RRID:MGI:3763368) (*Yagi et al., 2007*) and *Zp3-cre* (RRID:IMSR_JAX:003651) (*de Vries et al., 2000*) mouse lines were used in this study. Crosses to produce experimental embryos are described in figures. To obtain maternal/zygotic *Tead4* mutant embryos *Tead4 fl/fl, Zp3-cre* females were crossed to *Tead4 fl/-* males and genotype of embryos was determined by immunofluorescense staining against Tead4. Preimplantation embryos were flushed from oviducts or uteri with EmbryoMax M2 Medium (EMD Milipore, Etobicoke, Canada). Embryos were collected at appropriate time points from 5–8 week old hormone-primed (5 IU each pregnant mare serum gonadotropin (Sigma, Oakville, Canada) and human chorionic gonadotropin (Sigma, Oakville, Canada), 48 hr apart) and mated females, followed by careful staging based on morphology and number of Cdx2-eGFP positive cells (*Figure 1—figure supplement 1*). Zygotes were washed clean of cumulus cells by brief treatment with 300 µg/ml hyaluronidase (Sigma Oakville, Canada). If not immediately used, embryos were cultured in small drops of KSOM supplemented with amino acids (EMD Milipore) under mineral oil (Zenith Biotech, Guilford, CT) at 37°C, with 5% $CO_2$ for specified times. ROCK inhibitor (Y-27632, Sigma) was used at 50 µM concentration. Doxycycline (Sigma) was used at 1 µg/ml. All animal work was carried out following Canadian Council on Animal Care Guidelines for Use of Animals in Research and Laboratory Animal Care under protocols (protocol number: 20–0026H) approved by The Centre for Phenogenomics Animal Care Committee.

### Dissociation of embryos to single cells and quantification of Cdx2-eGFP

The zona pellucida was removed using acid Tyrode's solution (Sigma, Oakville, Canada) and embryos were washed in M2. Dissociation was performed by incubating embryos in TrypLE Select (Gibco, Thermo Fisher Scientific, Waltham, MA) for 3–6 min at 37°C followed by pipetting through fine pulled glass capillaries. Individual cells were picked, aligned in rows in M2 media under oil and imaged using a Zeiss Axiovert 200 inverted microscope equipped with a Hamamatsu C9100-13 EM-CCD camera, a Quorum spinning disk confocal scan head (Quorum Technologies Inc., Guelph, Canada) and Volocity acquisition software version 6.3.1 (Perkin Elmer, Santa Clara, CA). Cells were not used for experiments for 60 min after imaging to ensure image was not acquired during mitosis (*Yamagata and FitzHarris, 2013*). Cells dividing during this time were excluded from the analysis. Cdx2-eGFP was quantified by measuring average pixel intensities from single plane images of individual cells, focusing on the plane with maximum eGFP intensity. Average nuclear Cdx2-eGFP measurements were corrected for cytoplasmic background. The cut off between 'Cdx2-eGFP low' and 'Cdx2-eGFP high' was set at 500 fluorescent intensity units and was based on the clearly distinct Cdx2-eGFP low cell population at the 64 cell and >64 cell stages.

### Library preparation for single-cell RNA sequencing

Single-cells were directly dispensed in lysis buffer and cDNA libraries were generated using Smart-seq2 as previously described (*Petropoulos et al., 2016*; *Picelli et al., 2013*, *2014*).

## Single-cell RNA sequencing data pre-processing and quality control

Reads were mapped to the mouse genome (mm10) using STAR with default settings (RRID: SCR_004463) (*Dobin et al., 2013*) and only uniquely mapped reads were kept. Gene expression levels (RefSeq annotations) were estimated in terms of reads per kilobase exon model and per million mapped reads (RPKM) using our established pipeline, rpkmforgenes.py (*Ramsköld et al., 2009*). Read counts from regions where different RefSeq genes overlapped were excluded and cells with ~ ≥40% uniquely mapped reads were retained. Genes were filtered, keeping 15, 713 out of 24, 490 genes that were expressed in ≥2 cells with an expression cutoff of 1. Cells were quality-filtered based on Spearman's correlation, percent uniquely mapped (~ ≥40%) and the minimum number of expressed genes per cell (3500). Then PCA dimensionality reduction was conducted for each individual time point and additional outlier cells were identified. Batch or embryo effects were not observed in the dataset. Data for this study is available at NCBI Gene Expression Omnibus (RRID: SCR_007303; http://www.ncbi.nlm.nih.gov/geo/) under accession number GSE84892.

## Gene variability and temporal separation of cells using single-cell RNA sequencing data

This analysis was conducted as previously described (*Petropoulos et al., 2016*). Briefly, gene-variability statistic was calculated that adjusted for the mean-variance relationship present in single-cell RNA-sequencing data. This was done by assuming that the expression distribution of a gene follow a negative binomial for which the variance depends on the mean, $v = m + m^2/r$, where $r$ is the over-dispersion, implying that $cv^2 = v/m^2 = 1/m + 1/r$. To estimate the technical variability we fitted such a model to our ERCC spike-in read counts and a gene-variability statistic was then obtained by adjusting for the technical variability present when conditioning on the mean expression level (*Brennecke et al., 2013*). To determine the number of variable genes used, we tested 100, 250, 500 and 1000 of the most variable genes, and visually assessed clusters obtained using principal component analysis (PCA). Similar profiles were obtained regardless of input. Temporal separation of cells was obtained by applying dimensionality reduction technique, PCA. Similar profiles were obtained using t-Distributed Stochastic Neighbour Embedding (t-SNE), another dimensional reduction algorithm (data not shown) (*Hinton and van der Maaten, 2008*).

## Lineage segregation of cells using single-cell RNA sequencing data

Cells were stratified by the developmental time that they were collected and their corresponding Cdx2-eGFP values. Principal components of interest were identified by both observing a separation of developmental time and Cdx2-eGFP profile of the cells. Single-cell differential expression analysis (SCDE) (*Kharchenko et al., 2014*) was then performed between the 'ICM' (corresponding to the low Cdx2-eGFP group) and 'TE' (corresponding to the high Cdx2-eGFP) of the early 32 cells to determine lineage signatures for the ICM and TE. The top 50 genes obtained for both the ICM and TE were then used as input to determine the segregation of cells at the late and early 16 cell stage, by using Spearman's rank correlations. Following lineage classification, SCDE was then performed for all groups (developmental stage and lineage (ICM, TE and 'co-expressed')) for each time point. The SCDE algorithm requires non-normalized integer values, as such, the raw read counts were provided as input. Genes with zero reads across the samples being compared were discarded. Two-sided *p*-values were calculated from the Benjamini-Hochberg multiple testing corrected Z-score (cZ) using the normal distribution as null hypothesis, and a significance level of 0.05 was used to deem genes as significantly differentially expressed.

## Whole-mount immunofluorescence staining of embryos

Embryos were fixed in 4% formaldehyde at room temperature for 15 min, washed once in PBS containing 0.1% Tween-20 (PBS-T), permeabilized for 15 min in PBS 0.2% Triton X-100 and then blocked in PBS-T with 2% BSA (Sigma) and 5% normal donkey serum (Jackson ImmunoResearch Laboratories Inc., West Grove, PA) at room temperature for 2 hr, or overnight at 4°C. Primary and secondary antibodies were diluted in blocking solution, staining was performed at room temperature for ~2 hr or overnight at 4°C. Washes after primary and secondary antibodies were done three times in PBS-T. F-actin was stained using Alexa Flour 546-conjugated phalloidin (A22283, Life Technologies, Waltham, MA) diluted 1:200 and added during secondary antibody incubation. Embryos were mounted

in Vectashield containing Dapi (Vector Laboratories Canada Inc., Burlington, Canada) in wells of Secure Seal spacers (Molecular Probes, Thermo Fisher Scientific) and placed between two cover glasses for imaging. Primary antibodies: chicken anti-mCherry 1:600 (RRID: AB_2636881, NBP2-25158, Novus Biologicals, Littleton, CO); mouse anti-mCherry 1:500 (RRID: AB_2307319, 632543, Clontech, Takara Bio USA, Inc., Mountain View, CA, USA); chicken anti-GFP 1:400 (RRID: AB_2534023, A10262, Invitrogen, Thermo Fisher Scientific,); mouse anti-Tead4 1:100 (RRID: AB_2203086, sc-101184, Santa Cruz Biotechnology Inc., Mississauga, Canada); mouse anti-Yap 1:100 (RRID: AB_1131430, sc-101199, Santa Cruz Biotechnology Inc.); rabbit P-ezrin 1:200 (RRID: AB_330232, 3141S, Cell Signaling Technologies Inc., Danvers, MA); rabbit anti-Cdx2 1:600 (RRID: AB_1523334, ab76541, Abcam, Cambridge, United Kingdom); mouse anti-Cdx2 1:100 (RRID: AB_2335627, MU392-UC, Biogenex, Fremont, CA); goat anti-Sox2 1:100 (RRID: AB_355110, AF2018, RandD Systems, Minneapolis, MN); goat anti-Sox17 1:100 (RRID: AB_355060, AF1924, RandD Systems); rabbit anti-Gata4 1:100 (RRID: AB_2247396, sc-9053, Santa Cruz Biotechnologies Inc.); goat anti-Klf4 1:100 (RRID: AB_2130245, AF3158, RandD Systems); mouse anti-Rfp 1:100 (RRID: AB_1141717, ab65856, Abcam); mouse anti-Lrp2 1:100 (RRID: AB_1260798, NB110-96417, Novus Biologicals); mouse anti-Dab2 1:100 (RRID: AB_397837, 610464, BD Biosciences, San Jose, CA, USA); rat anti-Krt8 1:10 (RRID: AB_531826, TROMA-I antibody, Developmental Studies Hybridoma Bank, Iowa City, IA, USA); mouse anti Krt18 1:100 (RRID: AB_305647, ab668, Abcam). Secondary antibodies: (diluted 1:500) 448, 549 or 633 conjugated donkey anti-mouse, donkey anti-rabbit or donkey anti-goat DyLight (Jackson ImmunoResearch) or Alexa Fluor (Life Technologies).

## Confocal microscopy and image analysis

Images were acquired using a Zeiss Axiovert 200 inverted microscope equipped with a Hamamatsu C9100-13 EM-CCD camera, a Quorum spinning disk confocal scan head and Volocity aquisition software version 6.3.1 (RRID: SCR_002668). Single plane images or Z-stacks (at 1 μm intervals) were acquired with a 40x air (NA = 0.6) or a 20x air (NA = 0.7) objective. Images were analyzed using Volocity or Imaris software version 8.3 (RRID: SCR_007370, Bitplane, South Windsor, CT).

Time-lapse imaging was performed on the same microscope equipped with an environment controller (Chamlide, Live Cell Instrument, Seoul, South Korea). Embryos were placed in a glass-bottom dish (MatTek, Ashland, MA) in KSOM covered with mineral oil. A 20x air (NA = 0.7) objective lens was used. Images were taken every 20 min for 20 hr at 4 μm Z intervals.

Cdx2, eGFP and Yap measurements from fixed whole embryo specimens were quantified using the spot function of Imaris. Cdx2 and GFP intensities were normalized against Dapi, while for Yap the average nuclear intensity over average cytoplasmic intensity was calculated. Time-lapse movies were also analyzed using Imaris and average nuclear Cdx2-eGFP intensities were measured with the spot function.

## Single-cell aggregation to morula assay

ICR morulae (8 or 16 cell embryos) were used as host embryos. The zona pellucida was removed and embryos were washed in M2. Cdx2-eGFP was quantified in individual donor cells as described before. A single donor cell and a single host morula were then brought together in a micro-well generated by pressing a blunt end needle into the bottom of a plastic tissue culture dish (Falcon, Thermo Fisher Scientific) in drops of KSOM under oil. Such aggregation chimeras were cultured for 48 hr.

Number of chimeras generated per donor cell stage: 15 8 cell stage (one experiment), 63 early 16 cell stage (two experiments), 71 late 16 cell stage (two experiments): 74 early 32 cell stage (two experiments), 34 late 32 cell stage (three experiments), 23 64 cell stage (three experiments) and 13 > 64 cell stage (two experiments).

For time-lapse imaging host embryos were injected with membrane-RFP mRNA (see below) and single early 32 cell stage donor cells were introduced into the host embryo through a hole in the zona generated by a laser (XYRCOS, Hamilton Thorne Inc., Beverly, MA) using a micromanipulator (TransferMan NK2, Eppendorf Canada, Mississauga, Canada). Micromanipulations were performed in M2 under oil. Number of chimeras live imaged: five with Cdx2-eGFP low donor cells and six with Cdx2-eGFP high donor cells.

## mRNA synthesis and injection

mRNA was synthesized from pCS2 membrane-RFP plasmid (*Megason and Fraser, 2003*) using the mMESSAGE mMACHINE SP6 Kit (Invitrogen) and resuspended in RNase-free water. Microinjection was performed using a Leica microscope and micromanipulators (Leica Microsystems Inc., Richmond Hill, Canada). Injection pressure was provided by a FemtoJet (Eppendorf) and negative capacitance was generated using a Cyto721 intracellular amplifier (World Precision Instruments, Sarasota, FL, USA). Injections were performed in an open glass chamber in M2 medium. Both blastomeres of ICR 2 cell embryos were injected with 200 ng/μg mRNA. After injection embryos were cultured to the morula stage (8 to 16 cell) and used as host embryos in aggregation experiments, which were live imaged.

## Embryo reconstruction assay

Cdx2-eGFP was quantified in individual cells as described before, cells were grouped based on eGFP levels and re-scanned as a group to avoid errors. 16, 32 or 64 single cells (the same number as the original embryo) were brought together in a in a micro-well (same as in the single cell aggregation assay) in drops of media under oil.

Some 64 cell reconstructions were performed in an emptied zona pellucida: a mid blastocyst embryo was selected, a hole was made in the zona using a laser and the embryo was suctioned out of the zona using a micro-suction pipette. 64 single cells were then reintroduced into the zona through the hole. Embryos aggregated slightly better in emptied zonas than in micro-well; however no difference was observed on cell fate outcomes between methods.

KSOM was used to culture embryos made at 16 and most 32 cell stages, however we found that at the 64 cell stage using embryonic stem cell media (DMEM (Invitrogen), 2 mM GlutaMAX (Invitrogen), 0.1 mM 2-mercaptoethanol (Sigma), 0.1 mM MEM non-essential amino acids (Invitrogen), 1 mM sodium pyruvate (Invitrogen), 50 U/ml each penicillin/streptomycin (Invitrogen), 15% fetal bovine serum (Gibco), 1000 U/ml LIF (EMD Millipore)) greatly improved cell survival. We confirmed that fate outcomes were not affected by the choice of media, as reconstructions at the 32 cell stage in embryonic stem cell media gave the same results as using KSOM. Reconstructed embryos were cultured until embryonic day 4.5.

Number of reconstructed embryos generated for each developmental stage: 11 late 16 cell stage (two experiments), 19 early 32 cell stage (three experiments), 21 late 32 cell stage (four experiments) and 14 64 cell stage (five experiments).

## Generation of inducible dominant-negative Lats2 transgenic embryos

Pronuclear injections were performed on zygotes or cytoplasmic injections on 2 cell stage embryos. Typically only one blastomere of the 2 cell embryo was microinjected. ICR or *Cdx2-eGFP* (to aide embryo staging at the time of doxycycline addition) embryos were used. Microinjection was performed using a Leica microscope, micromanipulators (Leica Microsystems) and a FemtoJet (Eppendorf). A cocktail of two PiggyBAC (PB) plasmids and one mRNA was injected. The injection cocktail consisted of (i) PB-TAC-DNLats2-IRES-mCherry or PB-TAC-mCherry-IRES-mCherry (mCherry only control) (ii) PB-CAG-rtTA and (iii) PBase mRNA. For pronuclear injections (i) 5–10 ng/μl, (ii) 5–10 ng/μl and (iii) 20–40 ng/μl concentrations were used. For 2 cell cytoplasmic microinjections (i) 15 ng/μl, (ii) 15 ng/μl and (iii) 160 ng/μl concentrations were used. Transgenic embryos were collected using both microinjection methods with the same cell fate outcomes observed. However, we found that embryo development, as well as PiggyBAC integration efficiency was highly improved when 2 cell injections were used. Both microinjection methods typically generated mosaic embryos, in which mCherry positive (DN Lats2-mCherry transgenic) and mCherry negative (non-transgenic control) cells could be analyzed.

PB-TAC-DNLats2-IRES-mCherry: PB – PiggyBAC terminal repeat; TAC - tamoxifen activated minimal CMV promoter; DN Lats2 - dominant negative Lats2 (previously referred to as kdLats2 [*Nishioka et al., 2009*]). DN Lats2 was generously provided by Hiroshi Sasaki and was cloned into the PB-TAC IRES-mCherry construct, generously provided by Andras Nagy. PB-TAC mCherry-IRES-mCherry and PB-CAG-rtTA plasmids were also provided by Andras Nagy. For producing PBase mRNA, the PBase coding region was cloned into the pCS2+ vector. *In vitro* transcription was performed as described before.

Injected embryos were cultured to desired stage, imaged to ensure no leaky expression of DN Lats2-mCherry was present and induced with 1 µg/ml doxycycline (Sigma) for 24 hr. Transgenic embryos were identified by the presence of mCherry. A total of 42 DN Lats2-mCherry and 20 mCherry control transgenic embryos were made in eight independent experiments that had strong mCherry expression in inner cells.

## Statistics

No statistical method was used to predetermine sample size. For each experiment 10–15 females were used to harvest embryos. Pre-established criteria were used for all experiments to stage embryos (reported in *Figure 1—figure supplement 1*). Randomization was achieved by pooling all staged embryos in one experiment, with the exception of RNA sequencing experiments. For RNA sequencing experiments embryo information was preserved, however no embryo batch effect was noted when single cell expression profiles were analyzed. The investigator was not blinded to group allocation during experiments and outcome assessment, with the exception of RNA sequencing experiments. Cdx2-eGFP measurements of single cells and RNA sequencing of the same cells were performed by different investigators and outcomes were thus blindly determined.

Data were analyzed by two-sided *t*-test or one-way ANOVA (Graphpad Prism). Data were tested for normality of residuals (D'Agostino and Pearson omnibus normality test) and homogeneity of variances (F test or Bartlett's test) for each variable. When data failed to satisfy those premises a non-parametric test was chosen (Mann-Whitney or Kruskal-Wallis test).

For *Figure 6E* and *Figure 6—figure supplement 2* Chi-squared test was used to test the null hypothesis that cell fate is not affected by DN Lats2-IRES-mCherry or mCherry-IRES-mCherry expression. Expression of DN Lats2-IRES-mCherry or mCherry-IRES-mCherry was considered the independent variable and cell fate was the dependent variable.

## Comparison with Tead4 ChIP sequencing data

Significance of overlap between TE-specific genes (all stages of TE development considered, n = 404) and genes associated with Tead4 binding sites in mouse trophoblast stem cells (n = 8190, union of overlapping, 3' and 5' genes) as reported by *Home et al. (2012)* were determined using hypergeometric test, assuming 18,388 genes with experimentally-based functional annotations (MGI). *Figure 3D* was created using data from Home *et al.* overlapped with TE-specific genes in NAViGaTOR 3 (http://ophid.utoronto.ca/navigator) (RRID: SCR_008373) (*Brown et al., 2009*).

## Acknowledgements

This work is funded by the Canadian Institutes of Health Research (CHIR grant #FDN-143334) (JR), the Ragnar Söderberg Foundation, the Swedish Foundation for Strategic Research, the Knut and Alice Wallenberg Foundation and the Swedish Research Council (FL). EP was supported by the Restracomp fellowship from The Hospital for Sick Children and SP was supported by the Mats Sundin Fellowship in Developmental Health. We thank Kate McDole and Yixian Zheng for providing *Cdx2-eGFP* reporter mice, Hiroshi Sasaki and Andras Nagy for providing constructs and Amy Wong for useful comments on the manuscript. We thank the Transgenic Core lab at The Centre for Phenogenomics, Toronto for transgenic services and equipment use.

## Additional information

### Funding

| Funder | Grant reference number | Author |
| --- | --- | --- |
| Canadian Institutes of Health Research | FDN-143334 | Janet Rossant |
| Ragnar Söderbergs stiftelse | M67/13 | Fredrik Lanner |
| Stiftelsen för Strategisk Forskning | ICA12-0001 | Fredrik Lanner |
| Knut och Alice Wallenbergs | 2015.0096 | Fredrik Lanner |

| | | |
|---|---|---|
| Stiftelse | | |
| Vetenskapsrådet | 2013-2570 | Fredrik Lanner |
| Restracomp Fellowship | post-doctoral fellowship | Eszter Posfai |
| Mats Sundin Fellowship | post-doctoral fellowship | Sophie Petropoulos |

The funders had no role in study design, data collection and interpretation, or the decision to submit the work for publication.

#### Author contributions
EP, Conceptualization, Data curation, Formal analysis, Methodology, Writing—original draft, Writing—review and editing; SP, Data curation, Formal analysis, Writing—original draft; FROdB, JPS, Data curation; IJ, Formal analysis; RS, Supervision, Methodology; FL, Supervision, Funding acquisition, Methodology; JR, Conceptualization, Supervision, Funding acquisition, Writing—original draft, Writing—review and editing

#### Author ORCIDs
Janet Rossant, http://orcid.org/0000-0002-3731-5466

#### Ethics
Animal experimentation: All animal work was carried out following Canadian Council on Animal Care Guidelines for Use of Animals in Research and Laboratory Animal Care under protocols (protocol number: 20-0026H) approved by The Centre for Phenogenomics Animal Care Committee.

## Additional files

### Major datasets
The following dataset was generated:

| Author(s) | Year | Dataset title | Dataset URL | Database, license, and accessibility information |
|---|---|---|---|---|
| Posfai E, Petropoulos S, Barros F, Schell JP, Sandberg R, Lanner F, Rossant J | 2016 | Sequential loss of plasticity during trophectoderm and inner cell mass lineage segregation in the mouse embryo | https://www.ncbi.nlm.nih.gov/geo/query/acc.cgi?acc=GSE84892 | Publicly available at the NCBI Gene Expression Omnibus (accession no: GSE84892) |

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
