## [Decision Letter]

Thank you for submitting your article "Position- and Hippo signaling-dependent plasticity during lineage segregation in the early mouse embryo" for consideration by *eLife*. Your article has been reviewed by three peer reviewers, and the evaluation has been overseen by Robb Krumlauf as the Senior Editor and Reviewing Editor. The following individuals involved in review of your submission have agreed to reveal their identity: Hiroshi Sasaki (Reviewer #1); Anna-Katerina Hadjantonakis (Reviewer #2).

The reviewers have discussed the reviews with one another and the Reviewing Editor has drafted this decision to help you prepare a revised submission.

This is an important paper, which provides a comprehensive data set clarifying the relationships between global gene expression, lineage specification and commitment, and responsiveness to the Hippo signaling pathway during segregation of TE and ICM fates in preimplantation-stage mouse embryos. The performed experiments are of high quality and the results produced are novel and very important to the field. The provided comprehensive single-cell global transcriptional dataset, which covers the entire lineage segregation period with high temporal resolution, is a valuable resource. The importance of the paper resides not only in revealing the changes in global gene expression profiles during segregation of the cell fates, but also in experimentally linking these to the broader concepts of developmental biology, cell fate specification and commitment, and responsiveness to Hippo signaling, a key cell fate control signal. Although there has been a lot of work over the years to address these questions, this manuscript is the first to combine so many approaches to generate confirmatory data. The previous literature has generally been well incorporated and on the whole suitably attributed. Performing these assays in parallel resolved inconsistencies found in previously published literature. In conclusion, this is an excellent paper providing valuable information to the field. There is a request for inclusion of some additional experiments the reviewers would like to see addressed.

1) The experiments shown in Figure 6 are incomplete. First, the number of embryos examined is small. Only two to three embryos were examined at the early 32- to 64-cell stages, and only one embryo was examined for the >64-cell stage. Importantly, the two 64-cell stage embryos showed different responses. It is expected that such variability is present amongst embryos. Therefore, it is important to analyze a larger number of embryos, (e.g., >10 embryos, or at least five embryos per stage), to reach a convincing conclusion. This experiment also lacks appropriate controls: expression of Cdx2 and Sox2+Sox17 in inner cells of the normal embryos, and mCherry-only expressed embryos. Finally, quantitative results should be summarized in a graph with statistical analyses, as shown in other experiments, e.g., Figure 6.

---

## [Author Response]

[…] 1) The experiments shown in Figure 6 are incomplete. First, the number of embryos examined is small. Only two to three embryos were examined at the early 32- to 64-cell stages, and only one embryo was examined for the >64-cell stage. Importantly, the two 64-cell stage embryos showed different responses. It is expected that such variability is present amongst embryos. Therefore, it is important to analyze a larger number of embryos, (e.g., >10 embryos, or at least five embryos per stage), to reach a convincing conclusion. This experiment also lacks appropriate controls: expression of Cdx2 and Sox2^+^Sox17 in inner cells of the normal embryos, and mCherry-only expressed embryos. Finally, quantitative results should be summarized in a graph with statistical analyses, as shown in other experiments, e.g., Figure 6.

We have now analyzed a larger number of embryos and included additional controls (H_2_O-injected and mCherry only controls), as requested. We have also quantified the results and presented them in a graph with statistical analysis. Importantly, our conclusions from these additional data are essentially the same as our previous findings. Generation of inducible DN Lats2 transgenic embryos required the simultaneous integration of two PiggyBAC constructs. This proved to be an inefficient approach when we generated transgenic embryos by microinjecting the pronucleus of zygotes – hence the limited number of embryos analyzed previously. We now discovered that generating transgenic embryos by microinjecting into the cytoplasm at the 2-cell stage was markedly more efficient, allowing us to increase numbers of transgenic embryos.